# ONLINE MULTIMODAL LEARNING WITH HUMAN-IN-THE-LOOP

## ABSTRACT

We study the online multimodal learning (OML) problem, wherein a model is not frozen at any point in time but instead dynamically adapts its structure and parameters to learn new multimodal concepts and associations without forgetting the learned ones throughout its lifetime. To address this challenge, we propose a brain-inspired neural network with a hierarchical and modular architecture, named OML. Based on the characteristics of different hierarchies and modules, we design different types of artificial neuron models. The network includes ascending, descending, and lateral pathways, which ensure that all modalities can cooperate and interact with each other during online learning. Additionally, we develop a reference extraction algorithm that autonomously identifies the precise features to which a word refers. During online learning, the network performs conflict checking between the current input and the knowledge already learned from previous data. If a conflict occurs, the network is capable of posing appropriate questions to the user and updating itself based on the user's answers. All the designs make our method do learning like the way humans do. Experimental results demonstrate that our method can effectively handle the online multimodal learning.

## 1 INTRODUCTION

Humans can learn multimodal concepts continuously and interactively. For example, they can learn new words which name new objects throughout their lifetime; they can question the name of an object you teach which conflicts with their experience, then decide whether to learn it based on your answer. As mentioned in Kudithipudi et al. (2022) "**Humans learn from interactions with their environment throughout their lifetime.** To perceive the external environment, our brain uses multiple sources of sensory information derived from several different modalities, including vision, audition and taste. **All these sources of information are efficiently associated to form a coherent and robust percept. This is the cornerstone of human intelligence.**"

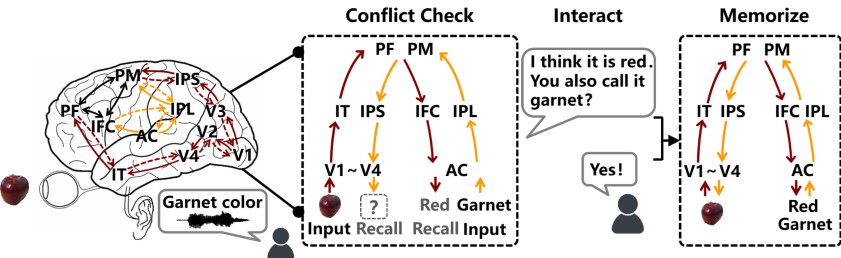

Figure 1: An example of online multimodal learning with human-in-the-loop. The red and yellow arrows indicate the signal flow of vision and audition in the brain.

Fig. 1 shows an example of online multimodal learning with human-in-the-loop. The teacher shows a red apple to the student and teaches the new color word "garnet". The student has previously learned a related word "red" but has not learned "garnet". Therefore, the student may ask, "I think it is red. You also call it garnet?" If the teacher gives a positive answer, e.g., "Yes!", the student memorizes this new word and its association with the object (apple). The process involves recognition, recall, conflict checking, interaction, and memorization, which is similar to the way humans learn.

In Srivastava & Salakhutdinov (2014), the authors propose the following view: A good multimodal learning model should learn representations which are useful for classification and retrieval, and be able to fill in missing modalities given observed ones. Furthermore, we believe that continuous learning capability and interactive ability are also indispensable. **Therefore, the multimodal learning model proposed in this work not only retains the characteristics outlined in** Srivastava & Salakhutdinov (2014), **but also incorporates the following additional attributes:**

**(1) It can continuously learn new multimodal concepts and new associations between the multimodal concepts in an online manner without forgetting the learned ones.**

**(2) It can detect conflict between the current input and the learned ones. If a conflict occurs, it can ask the user appropriate questions and conduct learning based on user's answer.**

## 2 RELATED WORK

Most multimodal learning methods can be divided into joint representation methods and coordinated representation methods Baltrusaitis et al. (2019). Joint representation methods usually introduce a joint layer over unimodal learning machines, which receives inputs from all modalities. Ngiam et al. (2011) train a restricted Boltzmann machine over two restricted Boltzmann machines which are pre-trained with visual and auditory samples. Srivastava & Salakhutdinov (2014) train a deep Boltzmann machine to jointly model word vectors and image features. Liu et al. (2022) project semantic representations of image and text modalities to a common Hamming space. Lin & Hu (2024) design a multimodal mixup network which consists of modality-specific and joint-modality encoders. The modality-specific encoders receive unimodal inputs and the joint-modality encoder receives input from all modalities. Sun et al. (2024) use Transformer-based encoder and decoder to learn a joint representation. He et al. (2025) design a multiscale fusion module which can integrate diverse components from different modalities and multiscale features. Coordinated representation methods learn the representation of each modality under specified constraints. Hu et al. (2019) minimize the intra-class variation of each data pair captured from two modalities of the same class. Chen et al. (2021) maximize cosine similarity between the feature vectors learned by a visual network and a text network. Jiang & Li (2021) design a modality-shared representation to learn a modality-exclusive representation. Xie et al. (2024) introduce a main semantics consistency loss to align the main semantics between two modalities. Li et al. (2024) design a cross-modal association probability composer which combines the distributions of image and word features. Wang et al. (2024) propose a dynamic noise separator to learn a coordinated representation in noisy environments. Duan et al. (2025) employ consistency learning to eliminate the cross-modal discrepancy.

Most of the above studies rarely pay attention to online learning, i.e., continuously learning new multimodal concepts without forgetting the learned ones. Recently, researchers have begun to study online multimodal learning. Xing et al. (2019; 2021) design an online learning network which creates new neurons and connections to learn and bind new multimodal concepts. Tan et al. (2019); Shubham et al. (2025) introduce a fusion adaptive resonance theory which generates new prototypes whose weights are set to the features of unrecognized multimodal patterns. However, these methods cannot learn precise references of concepts, detect conflicts or handle the conflicts through interaction with users, which are extremely important capabilities for online learning.

## 3 METHOD

As shown in Fig. 2, our network is a modular and hierarchical structure which includes the feature layer, the unimodal association layer and the multimodal association layer. The feature layer consists of feature neurons (FN) which respond to particular types of features (extracted by backbone networks), e.g., shape and color features in a visual channel, acoustic features in an auditory channel. We use $N^{\alpha_k}$ to denote the set of FNs of type $\alpha_k$, as shown in Fig. 2, $\alpha_k$ can be feature types $b$, $s$, $p$, etc. The unimodal association layer consists of unimodal association neurons (UAN) which associate different types of feature neurons to form unimodal concepts. For example, visual association neurons can associate shape feature neurons and color feature neurons to form visual concepts. Auditory association neurons can associate a series of syllable feature neurons to form words. We use $N^\beta$ to represent the set of UANs in channel $\beta$, as shown in Fig. 2, $\beta$ can be channels $V$, $A$, $X$, etc. The multimodal association layer consists of multimodal association neurons (MAN)

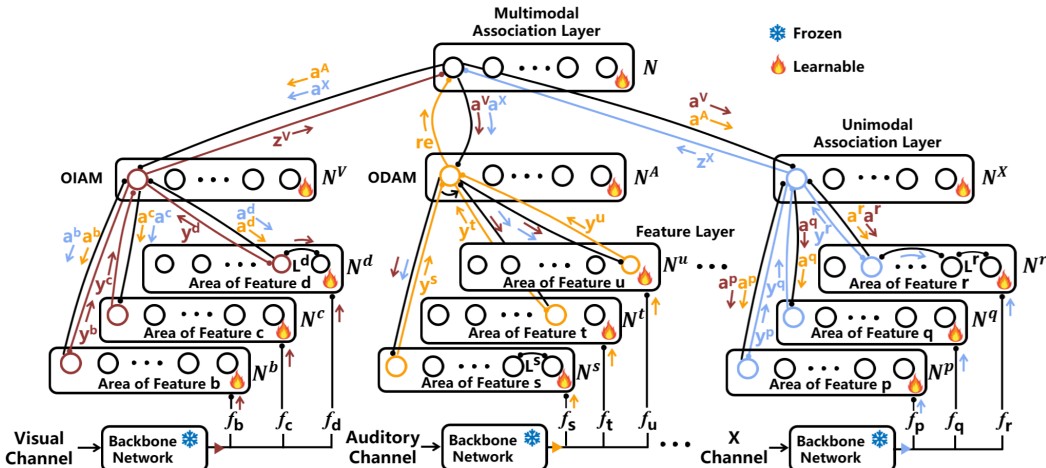

Figure 2: Structure of our network OML.

which associate unimodal association neurons in different channels. They transmit signals among channels and enable them to work together. We use $N$ to represent the set of MANs.

The network includes ascending, descending and lateral pathways. With these pathways, the network remembers the associations among concepts in different modalities, and the concepts in different modalities can activate each other. For example, as shown in Fig. 2, an input image can activate FNs, UANs and MANs via the ascending pathways in the visual channel. Then the activated MANs can activate UANs and FNs via the descending pathways in the auditory channel and the X channel. Lateral pathways exist between FNs with similar weights. During learning, activated FNs activate their similar FNs via the lateral pathways, which improves the generalization ability of the network.

Moreover, these pathways can enable the network to detect conflicts. For example, when a pair of image and word comes, if the input word is different from the words which are activated by the input image via these pathways, a conflict happens. In such a case, the network will ask the user questions for help. Meanwhile, the pathways can also help the auditory association neuron (word) locate the features to which it should refer. For example, the word "apple" should refer to combinations of some particular shapes and colors; the word "red" should refer to some particular color features.

New neurons and pathways can be added to the network during online learning when needed, e.g., when samples from new object or new association between samples arrive.

### 3.1 FEATURE NEURON

As shown in Fig. 2, a FN responds to some particular type of features. We use $N^{\alpha_k}$ to represent the set of FNs of type $\alpha_k$. Each neuron $N_j^{\alpha_k}$ has an ascending pathway and a descending pathway. The ascending pathway receives feature vector $\boldsymbol{x} = [x_1, x_2, ..., x_n]$. The ascending activation function $f_F^a$ of $N_j^{\alpha_k}$ is defined as follows,

$$f_F^a = \begin{cases} \boldsymbol{y}^{\alpha_k} = \sum_{i=1}^{n} \sum_{t=1}^{T} w_{j,i} \cos \lambda_i^{\alpha_k} 2\pi \frac{t-1}{T}, & d(\boldsymbol{x}, \boldsymbol{w}_j) \leq \theta \\ 0, & \text{otherwise} \end{cases} \quad (1)$$

where $\boldsymbol{w}_j = [w_{j,1}, w_{j,2}, ..., w_{j,n}]$ and $\theta$ are the weights and threshold of the FN $N_j^{\alpha_k}$. $d()$ is a distance function that measures the similarity between $\boldsymbol{x}$ and $\boldsymbol{w}_j$, we use Euclidean distance in practice. $\boldsymbol{y}^{\alpha_k}$ is an activation signal which will be transmitted to the unimodal association layer. $\lambda_i^{\alpha_k}$ is a frequency parameter which corresponds to the $i$-th dimension of the weights or features with the feature type $\alpha_k$. Here, each dimension corresponds to a unique frequency which means each feature type $\alpha_k$ corresponds to a unique frequency vector $\boldsymbol{\lambda}^{\alpha_k}$ in the network. We assign a unique natural number to each $\lambda_i^{\alpha_k}$ in practice. $T$ is a predefined parameter which is used to generate a period time of signal, its value does not affect the algorithm.

The descending pathways receive signals from UANs. $\boldsymbol{U}_{i,j}^{\alpha_k} = 1$ means there exists a descending connection from UAN $N_i^\beta$ to FN $N_j^{\alpha_k}$ (where feature area $\alpha_k$ is in channel $\beta$) and $\boldsymbol{U}_{i,j}^{\alpha_k} = 0$ means not. We use $\boldsymbol{a}^{\alpha_k} = [a_1^{\alpha_k}, a_2^{\alpha_k}, ..., a_m^{\alpha_k}]$ to represent a signal transmitted in a descending pathway between a UAN and a FN, and $\boldsymbol{A}^{\alpha_k} = [A_1^{\alpha_k}, A_2^{\alpha_k}, ..., A_m^{\alpha_k}]$ to denote the signal variable. Each dimension $A_i^{\alpha_k}$ corresponds to a frequency $\lambda$, which means this dimension receives an amplitude value $a_i^{\alpha_k}$ at frequency $\lambda$. $A_i^{\alpha_k}$ is modeled as a Gaussian distribution $A_i^{\alpha_k} \sim \boldsymbol{N}(\mu_i, \sigma_i)$, and a relative probability density of a sample $a_i^{\alpha_k}$ of $A_i^{\alpha_k}$ is calculated with

$$p_i^{\alpha_k} = \exp(-\frac{(a_i^{\alpha_k} - \mu_i)^2}{2\sigma_i^2}), \quad 1 \le i \le m$$

Then the descending activation function $f_F^d$ in this descending pathway is defined as follows,

$$f_F^d = \begin{cases} 1, & \forall p_i^{\alpha_k} \ge \vartheta, \ 1 \le i \le m \\ 0, & \text{otherwise} \end{cases} \tag{2}$$

where $\vartheta$ is the relative probability density threshold.

Lateral connections are established between feature neurons which have similar weights, in practice, feature neurons $N_i^{\alpha_k}$ and $N_j^{\alpha_k}$ satisfy $d(\boldsymbol{w}_i, \boldsymbol{w}_j) \le 2\theta$, where $\boldsymbol{w}_i$ and $\boldsymbol{w}_j$ are the weights of $N_i^{\alpha_k}$ and $N_j^{\alpha_k}$, $d()$ and $\theta$ are defined in Eq. (1). As shown in Fig. 2, we use a 0-1 matrix $\boldsymbol{L}^{\alpha_k}$ to represent the lateral connections. $L_{i,j}^{\alpha_k} = 1$ means there is a connection between neuron $N_i^{\alpha_k}$ and $N_j^{\alpha_k}$, $L_{i,j}^{\alpha_k} = 0$ means no connection. During online learning, the activated feature neurons can activate its laterally connected neurons.

## 3.2 Unimodal Association Neuron

A UAN connects different types of FNs to form a unimodal concept, e.g., to form a visual concept by connecting shape, color and other visual feature neurons, to form a word by connecting a group of syllable feature neurons in a specific order. We divide the activation mode of the UANs into two types: order independent activation mode (OIAM) and order dependent activation mode (ODAM). For example, visual association neurons have an OIAM, because different activation orders of shape, color and other feature neurons they connects do not affect their activation. Auditory association neurons have an ODAM, because the syllable feature neurons they connect must be activated in a specific order to form a correct word. We use $N^\beta$ to represent the set of UANs in channel $\beta$.

For the OIAM channel (e.g., a visual channel), ascending connections from FNs of type $\alpha_k$ to UANs are represented by a 0-1 matrix $\boldsymbol{W}^{\alpha_k}$, where $\boldsymbol{W}_{i,j}^{\alpha_k} = 1$ means there exists an ascending connection from FN $N_j^{\alpha_k}$ to UAN $N_i^\beta$ (e.g., $\beta = V$ for the visual channel) and $\boldsymbol{W}_{i,j}^{\alpha_k} = 0$ means not. Assuming there are $m$ different feature areas $\alpha_1, \alpha_2, ..., \alpha_m$ in channel $\beta$ (vision), the ascending activation function of the UAN $N_i^\beta$ is defined as follows,

$$f_U^a = \begin{cases} \boldsymbol{z}^\beta = \sum_{k=1}^m \boldsymbol{y}^{\alpha_k}, & \forall \boldsymbol{W}_{i,:}^{\alpha_k} \cdot \boldsymbol{e}^{\alpha_k} = 1 \\ 0, & \text{otherwise} \end{cases} \tag{3}$$

where $\boldsymbol{W}_{i,:}^{\alpha_k}$ is the $i$-th row of $\boldsymbol{W}^{\alpha_k}$, $\boldsymbol{e}^{\alpha_k}$ is a 0-1 vector, $e_j^{\alpha_k} = 1$ if the FN $N_j^{\alpha_k}$ is activated. $\boldsymbol{y}^{\alpha_k}$ represents the signals generated by the activated FN with feature type $\alpha_k$ in channel $\beta$ using Eq. (1). $\boldsymbol{z}^\beta$ is the activation signal which equals the sum of the signals of FNs to which the UAN connects.

The descending pathways receive signals from MANs. $\boldsymbol{U}_{j,i}^\beta = 1$ means there exists a descending connection from MAN $N_j$ to UAN $N_i^\beta$ and $\boldsymbol{U}_{j,i}^\beta = 0$ means not. We use $\boldsymbol{a}^\beta = [a_1^\beta, a_2^\beta, ..., a_s^\beta]$ to represent a signal transmitted in a descending pathway, and $\boldsymbol{A}^\beta = [A_1^\beta, A_2^\beta, ..., A_s^\beta]$ to denote the signal variable. Each dimension $A_i^\beta$ corresponds to a frequency $\lambda$, which means this dimension receives an amplitude value $a_i^\beta$ at frequency $\lambda$. The descending activation function is modeled similarly to Eq. (2),

$$f_U^d = \begin{cases} \boldsymbol{a}^{\alpha_k}, & \forall p_i^\beta \ge \vartheta, \ 1 \le i \le s \\ 0, & \text{otherwise} \end{cases} \tag{4}$$

where $\boldsymbol{a}^{\alpha_k}$ is a descending signal which is transmitted to feature area $\alpha_k$.

For the ODAM channel (e.g., an auditory channel), ascending connections from FNs with type $\alpha_k$ to UANs are represented by a 3-D 0-1 matrix $\boldsymbol{W}^{\alpha_k}$. $\boldsymbol{W}^{\alpha_t}_{t,i,j} = 1$ means there exists a connection between UAN $N_i^\beta$ (e.g., $\beta = A$ for the auditory channel) and FN $N_j^{\alpha_t}$ at position $t$ of the feature neuron series to which the UAN $N_i^\beta$ connects, $\boldsymbol{W}^{\alpha_t}_{t,i,j} = 0$ means not. The ascending activation function of $N_i^\beta$ is defined as follows,

$$f_U^a = \begin{cases} \boldsymbol{z}^\beta = re(\boldsymbol{\mu}, \boldsymbol{\sigma}), & \forall \boldsymbol{W}^{\alpha_t}_{t,i,:} \cdot \boldsymbol{e}^{\alpha_t} = 1 \\ 0, & \text{otherwise} \end{cases} \tag{5}$$

where $\boldsymbol{W}^{\alpha_t}_{t,i,:}$ is the $i$-th row of $\boldsymbol{W}^{\alpha_t}$ at position $t$. $\boldsymbol{e}^{\alpha_t}$ is a 0-1 vector, $e_j^{\alpha_t} = 1$ if FN $N_j^{\alpha_t}$ is activated at position $t$. $\boldsymbol{z}^\beta$ is the output signal, where $re()$ is a reference extraction function that finds some particular parts of features to which the neuron refers. The details will be introduced in Section 3.4. The descending activation function is modeled similarly to Eq. (4).

## 3.3 Multimodal Association Neuron

As shown in Fig. 2, MANs connect UANs in different channels. They transmit signals from one channel to other channels and enable different channels to work together. We use $N$ to represent the set of MANs.

Ascending connections from UANs in channel $\beta$ to MANs are represented by a 0-1 matrix $\boldsymbol{W}^\beta$, where $\boldsymbol{W}^\beta_{i,j} = 1$ means there exists an ascending connection from UAN $N_j^\beta$ to MAN $N_i$ and $\boldsymbol{W}^\beta_{i,j} = 0$ means not. The ascending activation function of the MAN $N_i$ is defined as follows,

$$f_M^a = \begin{cases} [\boldsymbol{a}, \boldsymbol{\lambda}] = \mathcal{F}(\boldsymbol{z}^\beta), & \boldsymbol{W}^\beta_{i,:} \cdot \boldsymbol{e}^\beta = 1 \\ 0, & \text{otherwise} \end{cases} \tag{6}$$

where $\boldsymbol{W}^\beta_{i,:}$ is the $i$-th row of $\boldsymbol{W}^\beta$. $\boldsymbol{e}^\beta$ is a 0-1 vector, $e_j^\beta = 1$ if the UAN $N_j^\beta$ is activated. $\boldsymbol{z}^\beta$ is the output of the activated UAN in channel $\beta$, $\mathcal{F}()$ is the Fourier transform. The output $[\boldsymbol{a}, \boldsymbol{\lambda}]$ are the amplitude and frequency obtained by $\mathcal{F}()$, for convenient, we write $\boldsymbol{a}^\beta = [\boldsymbol{a}, \boldsymbol{\lambda}]$ . The amplitude $\boldsymbol{a}$ can be transmitted to all the other channels via descending connections according to their signal variable $\boldsymbol{A}^\beta$ and frequency $\boldsymbol{\lambda}$ (for finding the correct pathways) attached to $\boldsymbol{A}^\beta$.

## 3.4 Reference Extraction

**How does a learner pick out the correct part of a visual signal to which a word refers?** Fig. 3(a) shows an example scenario where the network is taught to understand the word "hóng sè" (red in English). The network is fed with images of red onions and apples and word "hóng sè". The word neuron receives visual channel's signals $\boldsymbol{a}^{V,t} = [\boldsymbol{a}^{b,t}, \boldsymbol{a}^{c,t}]$ which are generated by Eq. (6), Eq. (3) and Eq. (1). According to these equations, $\boldsymbol{a}^{b,t}$ and $\boldsymbol{a}^{c,t}$ are the shape (e.g., assumed to be area of feature $b$ in Fig. 2) and color (e.g., assumed to be area of feature $c$ in Fig. 2) features. From the two objects, it can be found that the values of color features are more stable than those of shape features.

To measure stability, we introduce a reference extraction function. During online learning, the mean and variance of the signals $\boldsymbol{a}^{V,t}$ are calculated,

$$\boldsymbol{\mu} = \sum_{t=1}^n \boldsymbol{a}^{V,t}/n = [\boldsymbol{\mu}^b, \boldsymbol{\mu}^c], \qquad \boldsymbol{\sigma} = \sum_{t=1}^n (\boldsymbol{a}^{V,t} - \boldsymbol{\mu}) \circ (\boldsymbol{a}^{V,t} - \boldsymbol{\mu})/n = [\boldsymbol{\sigma}^b, \boldsymbol{\sigma}^c]$$

where $\circ$ is the Hadamard product. If the word "hóng sè" actually refers to a visual concept, the variance of some dimensions in the visual signal must shrink and stabilize after sufficient samples have been learned. We use $c$ to represent such dimensions, then $\boldsymbol{\sigma}^c$ should be small. The variance of the other dimensions $b$ (e.g., dimensions for shape features) should increase and finally be much larger than those of the dimensions $c$. Finally the auditory association neuron can determine that the word "hóng sè" refers to the mean value of dimensions $c$. To pick out the referring dimensions, the coefficient of variation of each dimension is calculated,

$$\boldsymbol{r} = \boldsymbol{\sigma} \oslash \boldsymbol{\mu}$$

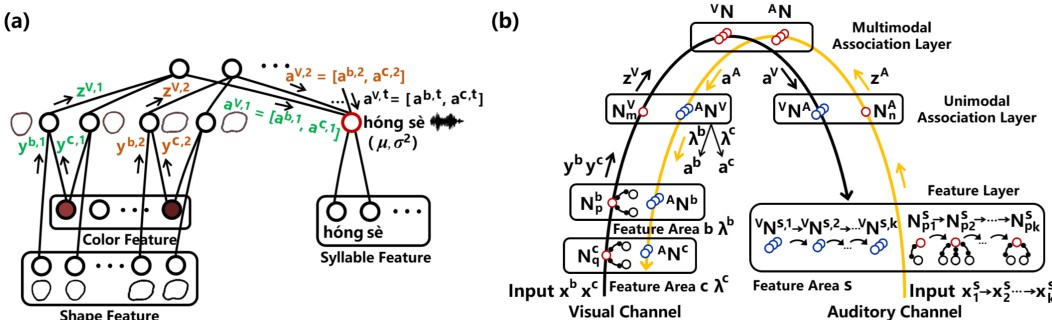

Figure 3: (a) An auditory association neuron refers to the Chinese word "hóng sè" which means red in English. It receives a series of visual channel's signals, $\boldsymbol{a}^{V,t} = [\boldsymbol{a}^{b,t}, \boldsymbol{a}^{c,t}]$ at time $t$, where $\boldsymbol{a}^{b,t}$ and $\boldsymbol{a}^{c,t}$ are shape and color features. (b) Ascending and descending activations in the visual and auditory channels. Red and blue circles represent the activated neurons in the ascending and descending pathways respectively. Black circles represent the neurons which are connected to the activated neurons by lateral connections.

where $\oslash$ is the element-wise division. If the word neuron refers to some particular parts of features, the corresponding coefficient of variation should be small.

For the visual channel (other channels are processed similarly), we assume there are $m$ different types of features. Correspondingly, $\boldsymbol{r}$ is divided into $m$ parts,

$$\boldsymbol{r} = [\boldsymbol{r}^{\alpha_1}, \boldsymbol{r}^{\alpha_2}, ..., \boldsymbol{r}^{\alpha_m}]$$

Then we get the maximum value $r'_j$ from each $\boldsymbol{r}^{\alpha_j}$

$$\boldsymbol{r}' = [\max(\boldsymbol{r}^{\alpha_1}), \max(\boldsymbol{r}^{\alpha_2}), ..., \max(\boldsymbol{r}^{\alpha_m})] = [r'_1, r'_2, ..., r'_m]$$

If $r'_j$ is larger than a threshold $r$, the word neuron does not refer to feature type $\alpha_j$. Finally, the features which are most likely the word neuron refers to are picked out.

The reference extraction $re()$ in the ascending activation function Eq. (5) is defined as follows,

$$re(\boldsymbol{\mu}, \boldsymbol{\sigma}) = [H(r'_1 - r), H(r'_2 - r), ..., H(r'_m - r)] \tag{7}$$

where $H(r'_j - r) = 1$ when $r'_j - r \leq 0$, which means feature type $\alpha_j$ is picked out, and $H(r'_j - r) = 0$ otherwise.

### 3.5 LEARNING WITH HUMAN IN THE LOOP

To learn multimodal concepts and associations, it is better to input a pair of samples. Here, for convenience, a pair of an image (OIAM channel) and a word (ODAM channel) about an object is used to describe the learning process. As shown in Fig. 3(b), when the network receives a pair of image and word, it first extracts visual features from the image and acoustic features from the word using respective feature extraction backbone networks. Assume that we get features $\boldsymbol{x}^b$ and $\boldsymbol{x}^c$ in the visual channel and $\boldsymbol{x}^s_1, \boldsymbol{x}^s_2, ..., \boldsymbol{x}^s_k$ ($k$ syllables in the word) in the auditory channel. The features are transmitted to their corresponding feature areas according to their feature types. Ascending and descending activations are then executed by the visual channel and auditory channel with functions Eq. (1) to Eq. (6). The ascending and descending activations in the two channel can form four combinations.

(1) The visual channel does not recognize the current image and the auditory channel recognizes the current word. In this scenario, the ascending activations in the visual channel do not occur. We initialize new FNs, e.g., $N_p^b$ and $N_q^c$, whose weights are initially set to the corresponding features extracted from the image. A new UAN $N_m^V$ and ascending connections are also initialized to associate the FNs to form a visual concept, i.e., $\boldsymbol{W}_{m,p}^b = 1$ and $\boldsymbol{W}_{m,q}^c = 1$.

The ascending activations in the auditory channel and descending activations in the visual channel which are launched by the auditory channel are executed as the yellow arrow in Fig. 3(b) shows.

Assume that sets $^AN^b$ and $^AN^c$ represent the visual FNs which are activated by the auditory channel. We denote $N_p^b$ and its laterally connected neurons as $G_p^b$, $N_q^c$ and its laterally connected neurons as $G_q^c$. If $^AN^b \cap G_p^b \neq \varnothing$ and $^AN^c \cap G_q^c \neq \varnothing$, this means the visual concepts activated by the auditory input are similar to the visual input. The current visual-auditory input pair is consistent with some previous ones. We add $N_m^V$ to the descending pathway of MANs in set $^AN$ which are activated by the current auditory input, i.e., set the descending connection matrix $\boldsymbol{U}_{i,m}^A = 1$, where $N_i \in {}^AN$. If $^AN^b \cap G_p^b = \varnothing$ or $^AN^c \cap G_q^c = \varnothing$, the visual concepts activated by the auditory input are not similar to the visual input. A conflict occurs. The network asks a question: "The object I recalled with the current auditory input does not look like the current visual input, are you sure to name the current visual input using the current auditory input?" If the user inputs a positive answer, e.g., "yes", $N_m^V$ is added to the descending pathway of MANs in set $^AN$ as the above operations. The mean $\boldsymbol{\mu}$ and variance $\boldsymbol{\sigma}$ of the activated word neuron $N_n^A$ are updated incrementally as follows,

$$\boldsymbol{\sigma} = \frac{(t-1)(\boldsymbol{\sigma} + \boldsymbol{\mu} \circ \boldsymbol{\mu}) + \boldsymbol{a}^V \circ \boldsymbol{a}^V}{t} - \frac{((t-1)\boldsymbol{\mu} + \boldsymbol{a}^V) \circ ((t-1)\boldsymbol{\mu} + \boldsymbol{a}^V)}{t^2}, \qquad \boldsymbol{\mu} = \frac{(t-1)\boldsymbol{\mu} + \boldsymbol{a}^V}{t} \qquad (8)$$

where $\boldsymbol{a}^V$ is generated by Eq. (6), $t$ is the total number of signals which $N_n^A$ has handled. If the user inputs a negative answer, e.g., "no", $N_m^V$ is not added to the descending pathway.

(2) The visual channel recognizes the current image and the auditory channel does not recognize the current input word. We initialize new FNs, $N_{p_1}^s$, $N_{p_2}^s$, ..., $N_{p_k}^s$ whose weights are initially set to the corresponding features extracted from each syllable. A new UAN $N_n^A$ and ascending connections are also initialized to associate the FNs to form an auditory concept, i.e., $\boldsymbol{W}_{1,n,p_1}^s = 1, ..., \boldsymbol{W}_{k,n,p_k}^s = 1$.

The ascending activations in the visual channel and descending activations in the auditory channel which are triggered by the visual channel are executed as the black arrow in Fig. 3(b) shows. Assume that sets $^VN^{s,1}$, $^VN^{s,2}$, ..., $^VN^{s,k}$ represent the auditory FNs which are activated by the visual input. We denote $N_{p_i}^s$ and its laterally connected neurons as $G_{p_i}^s$ $(1 \leq i \leq k)$. If all $^VN^{s,i} \cap G_{p_i}^s \neq \varnothing$ $(1 \leq i \leq k)$, this means some auditory concepts activated by the current visual input are similar to the current auditory input. The current visual-auditory input pair is consistent with previous ones. We add $N_n^A$ to the descending pathway of MANs in set $^VN$ which are activated by the current visual input, i.e., set the descending connection matrix $\boldsymbol{U}_{i,n}^V = 1$, where $N_i \in {}^VN$. If any $^VN^{s,i} \cap G_{p_i}^s = \varnothing$ $(1 \leq i \leq k)$, the auditory concepts activated by the visual input are not similar to the auditory input. A conflict occurs. The network picks out a name $^VN_i^A$ from set $^VN^A$ and asks a question, "I think you call it $^VN_i^A$ before, now you call it also $N_n^A$?" If the user inputs a positive answer, $N_n^A$ is added to the descending pathway of MANs in set $^VN$ as the above operations. If the user inputs a negative answer, $N_n^A$ is not added to the descending pathway.

(3) Both the visual and auditory channels recognize the current input. The ascending and descending activations in both channels are executed as the black and yellow arrows in Fig. 3(b) show. $^VN$ and $^AN$ represent the MANs activated by the visual and auditory channels. If $^AN \cap {}^VN \neq \varnothing$, the current input pair is consistent with some previously learned ones. $N_n^A$ is added to the descending pathway of the MANs in set $^VN$. The $\boldsymbol{\mu}$ and $\boldsymbol{\sigma}$ of $N_n^A$ are updated with Eq. (8). If $^AN \cap {}^VN = \varnothing$, reference extraction $re()$ is performed to $N_n^A$ and each neuron $^VN_i^A$ in set $^VN^A$. Then the network selects a neuron $^VN_i^A$ whose referring is same with that of $N_n^A$ and asks a question, "You call it $^VN_i^A$ before, now you also call it $N_n^A$?" If the user inputs a positive answer, $N_n^A$ is added to the descending pathway of the MANs in set $^VN$. If the user inputs a negative answer, $N_n^A$ is not added to the descending pathway.

(4) Neither the visual channel nor the auditory channel recognizes the current input. A new MAN $N_i$ and corresponding ascending and descending pathways are added to the network to associate $N_m^V$ and $N_n^A$, i.e., $\boldsymbol{W}_{i,m}^V = 1, \boldsymbol{W}_{i,n}^A = 1, \boldsymbol{U}_{i,m}^V = 1, \boldsymbol{U}_{i,n}^A = 1$.

## 4 EXPERIMENTS

We use the datasets used in Xing et al. (2019) and Lai et al. (2011). The dataset Xing et al. (2019) contains images and uttered Chinese names of common fruits. We denote this dataset as **Fruits**. The dataset Lai et al. (2011) contains images of objects that are commonly found in home environments. We take images of fruit objects from it and pair them with the voice data from Xing et al. (2019). We denote this dataset as **HomeF**. The experiment on the two datasets serve as the baseline experiment.

Table 1: Results on the Fruits and HomeF datasets. Best results are in bold. V: Vision, A: Audition.

| Dataset | Environment | Task | Offline Methods | | | | | Online Methods | | |
|---------|-------------|------|------|------|-------|------|------|------|------|------|
| | | | DAE | DBM | DJSRH | NRCH | FUME | ART | AEN | OML |
| **Fruits** | Close | V → A | 67.0 | 70.5 | 91.8 | **92.3** | 92.1 | 82.7 | 85.1 | 89.2 |
| | | A → V | 59.4 | 55.7 | 92.1 | 92.5 | **92.7** | 82.2 | 84.0 | 88.7 |
| | Open | V → A | 52.3 | 54.3 | 83.1 | 86.5 | 85.9 | 84.2 | 86.2 | **89.8** |
| | | A → V | 41.0 | 42.9 | 86.3 | 84.4 | 84.8 | 83.0 | 84.9 | **89.0** |
| **HomeF** | Close | V → A | 63.8 | 64.3 | 88.9 | **89.8** | 89.4 | 80.1 | 81.2 | 85.0 |
| | | A → V | 56.3 | 57.5 | 85.7 | 86.2 | **86.5** | 77.9 | 79.1 | 82.9 |
| | Open | V → A | 49.2 | 51.0 | 76.1 | 78.4 | 77.5 | 80.8 | 82.3 | **85.5** |
| | | A → V | 45.6 | 43.3 | 73.4 | 76.9 | 76.0 | 78.6 | 80.4 | **83.6** |

Table 2: Experimental results on the E-Fruits and E-HomeF datasets. Significant drops in accuracy compared with the corresponding results in the baseline experiment (Table 1) are marked by ↓.

| Dataset | Environment | Task | Offline Methods | | | | | Online Methods | | |
|---------|-------------|------|------|------|-------|------|------|------|------|------|
| | | | DAE | DBM | DJSRH | NRCH | FUME | ART | AEN | OML |
| **E-Fruits** | Close | V → A | 60.7$^{\downarrow}$ | 62.5$^{\downarrow}$ | 78.4$^{\downarrow}$ | 81.6$^{\downarrow}$ | 82.7$^{\downarrow}$ | 80.8 | 82.9 | **87.3** |
| | | A → V | 48.5$^{\downarrow}$ | 47.8$^{\downarrow}$ | 81.8$^{\downarrow}$ | 84.1$^{\downarrow}$ | 83.8$^{\downarrow}$ | 79.3 | 81.1 | **85.9** |
| | Open | V → A | 44.6$^{\downarrow}$ | 48.3$^{\downarrow}$ | 75.9$^{\downarrow}$ | 75.0$^{\downarrow}$ | 76.3$^{\downarrow}$ | 82.2 | 84.1 | **87.8** |
| | | A → V | 37.0 | 39.5 | 78.2$^{\downarrow}$ | 74.7$^{\downarrow}$ | 75.8$^{\downarrow}$ | 81.8 | 82.5 | **86.2** |
| **E-HomeF** | Close | V → A | 57.4$^{\downarrow}$ | 58.5$^{\downarrow}$ | 75.8$^{\downarrow}$ | 76.3$^{\downarrow}$ | 77.1$^{\downarrow}$ | 78.5 | 80.3 | **82.7** |
| | | A → V | 51.9 | 53.2 | 72.7$^{\downarrow}$ | 74.0$^{\downarrow}$ | 73.5$^{\downarrow}$ | 76.6 | 78.4 | **81.2** |
| | Open | V → A | 41.3$^{\downarrow}$ | 45.2$^{\downarrow}$ | 68.0$^{\downarrow}$ | 70.0$^{\downarrow}$ | 69.4$^{\downarrow}$ | 79.3 | 80.7 | **83.3** |
| | | A → V | 37.8$^{\downarrow}$ | 37.5$^{\downarrow}$ | 66.2$^{\downarrow}$ | 69.5$^{\downarrow}$ | 71.1$^{\downarrow}$ | 77.4 | 79.5 | **82.9** |

To verify the referring algorithm, we add color-referring uttered Chinese words to **Fruits** and **Home-F** to make two enhanced datasets, **E-Fruits** and **E-HomeF**. **To test the continuous learning ability, we use the learned networks from the baseline experiment to continue learning the two enhanced datasets.**

Another important ability for online learning is model reuse for extension of new modalities. Xing et al. (2021) build a model AEN which integrates a sudden emerged new input channel in an online way. **Following them, we extend our trained visual-auditory network above with a taste channel to continue learning taste concepts.** The dataset in Xing et al. (2021) contains taste data. Here, we add Chinese words referring to taste to the dataset, e.g., word "tián" and "suān" (sweet and sour in English). We denote this dataset as **VAT**. We also do the same augmentation on the dataset in Lai et al. (2011) which is denoted as **VAT-HomeF**.

We conduct the experiment in close and open environments. In the close environment, samples are randomly chosen from the whole dataset. In the open environment, we divide the dataset into four equal parts, each containing different classes. We first feed one part to the network. After learning is completed, we feed the next part and so forth. The open environment is designed to verify whether the network can handle the **catastrophic forgetting problem**. The parameters are set as follows: In Eq. (1), $\theta$ of the feature neuron is set to a quarter of the 2-norm of the weight of the neuron and $T$ is set to 150. In Eq. (2) and Eq. (4), $\vartheta$ is set to 0.8 which means a relative probability of 80%, $r$ in (7) is set to 0.5. For the visual channel, the backbone is the SAM Kirillov et al. (2023) which extracts objects from images, then we calculate the normalized Fourier descriptor of the object boundary as the shape feature, the mean value of the color inside the boundary as the color feature. For the auditory channel, the short-time energy and short-time zero-crossing rate are used to extract each syllable contained in the sample, and then the MFCCs of each syllable are extracted as auditory features. For the taste channel, the features are the taste features provided in Xing et al. (2021). We compare our method OML with DAE Ngiam et al. (2011), DBM Srivastava & Salakhutdinov (2014), DJSRH Su et al. (2019), NRCH Wang et al. (2024), FUME Duan et al. (2025), ART Shubham et al. (2025) and AEN Xing et al. (2021). DAE and DBM learn multimodal joint representations. DJSRH, NRCH, and FUME learn multimodal coordinated representations. These five methods are offline paradigms, they can be iteratively optimized multiple times on the dataset and the model is frozen after training. ART, AEN, and OLM learn multimodal representations in an online manner, and they

Table 3: Experimental results on the VAT and VAT-HomeF. V: Vision, A: Audition, T: Taste.

| Dataset | Environment | Method | T→V | T→A | V→A | V→T | A→V | A→T |
|---------|-------------|--------|-----|-----|-----|-----|-----|-----|
| **VAT** | Close | AEN | 88.3 | 87.4 | 82.0 | 86.6 | 80.7 | 86.4 |
| | | OML | **90.1** | **91.7** | **86.6** | **91.2** | **85.0** | **90.9** |
| | Open | AEN | 89.2 | 89.0 | 83.6 | 87.3 | 81.9 | 86.7 |
| | | OML | **92.1** | **93.9** | **87.2** | **91.7** | **85.8** | **91.8** |
| **VAT-HomeF** | Close | AEN | 80.6 | 80.2 | 78.5 | 79.1 | 76.8 | 79.3 |
| | | OML | **83.9** | **83.5** | **81.8** | **83.6** | **80.7** | **82.5** |
| | Open | AEN | 81.3 | 80.9 | 79.5 | 80.2 | 78.7 | 79.9 |
| | | OML | **84.2** | **83.8** | **82.6** | **84.0** | **82.1** | **82.8** |

learn each sample in the dataset only once. In the experiment, if the question posed to the user by OLM remains unanswered for a certain period of time, we set the answer to be positive.

### 4.1 PERFORMANCE COMPARISONS

To test the learned network, we use one channel input to get outputs from other channels on the testing dataset. For example, we use an image to recall its name and the words describing its color.

**(1) Baseline Experiment:** Table 1 shows the results. Compared with the online methods, OML gets the highest accuracy. The accuracy of OML is slightly lower than the offline methods in the close environment. In the open environment, the accuracy of the offline methods drops significantly due to the catastrophic forgetting, while OML is stable and achieves the highest accuracy.

**(2) Precise Referring Experiment:** Table 2 shows the results. OML gets the highest accuracy in both environments. The accuracy of the offline methods drops significantly compared with the baseline experiment, marked by ↓. This result is caused by the fact that continuous learning of novel color words disrupts previously learned knowledge. ART and AEN can learn the binding of image-name and image-color. However, they cannot learn a precise referring of a word. For example, they treat the name words and color words without difference. They cannot learn that the color word refers to an attribute of an object, i.e., a part of the feature vector of the object; the name word usually refers to all the attributes which form the object. As a result, when we use word "hóng sè" (red) to do recalling, they return all features (shape and color) of red objects (we count this as a correct result for them in Table 2). OML can effectively handle these problems, e.g., it can learn to find different referring patterns of the name and color words as Fig. 3(a) shows.

**(3) Modal Extension Experiment:** Table 3 shows the results. Because only AEN deals with the modal extension problem, we compare our method with AEN. Our method gets better results than AEN. Notably, AEN cannot distinguish whether a word refers to a taste or visual concept, e.g., "tián" (sweet) for a taste concept and "hóng sè" (red) for a visual concept. Because when it learns the image-word and taste-word pairs, it just associates them without distinction. As a result, when we perform recall using word "tián", AEN returns concepts in both the visual and taste channels (we count this as a correct result for AEN in Table 3), the same for the word "hóng sè". While OML binds them with the help of the $\lambda$ parameter. During recall, the signal can find its correct descending pathways by matching the $\lambda$ parameter (as shown in Fig. 3(b)) and activate the concept to which it exactly refers, i.e., "tián" and "hóng sè" each recall information from the taste channel and visual channel, respectively. Moreover, when we randomly add 10% of word-image or word-taste data pairs with incorrect matches, OML is able to detect all conflicts and raise appropriate questions.

## 5 CONCLUSION

We propose an online multimodal learning network, which is a hierarchical modular structure with ascending, descending and lateral pathways. The concept referring algorithm can autonomously locate the precise features to which a word refers. Conflict detection is executed every time the network receives a learning sample. When a conflict occurs, the network will ask user appropriate questions and do learning based on the user's answer. All these designs enable our method to learn in a manner similar to humans. In the experiment, we designed extensive experiments to verify the effectiveness of our method. Experimental results validate the characteristics we have claimed.

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
