# OpenReview forum: "Online Multimodal Learning with Human-in-the-Loop"
_ICLR.cc/2026/Conference — ICLR 2026 Conference Withdrawn Submission_

### Official Review · Reviewer_o2ai · 2025-10-26

**Soundness:** 2
**Presentation:** 2
**Contribution:** 3
**Rating:** 2
**Confidence:** 3

**Summary:**

This paper proposes a multimodal network architecture capable of continuous learning. It achieves continuous learning through dynamic adjustments to the architecture and avoids forgetting existing knowledge when acquiring new knowledge. The proposed method is built on three network layers: Feature Neurons (FN) for handle distinct features of each modality; Unimodal Association Neurons (UAN) for enable the understanding of unimodal information concepts; and Multimodal Association Neurons (MAN) for synchronize cross-modal information. The architecture distinguishes and recognizes different concepts through bidirectional pathways between the three layers and lateral pathways within the same layer. During the continuous learning process, the network is continuously refined to address four typical human-in-the-loop conflict scenarios, thereby realizing online learning.

**Strengths:**

1.The network is rain-Inspired and have Well-Designed Architecture. Drawing inspiration from how the human brain processes multimodal information, the paper designs a three-layer network architecture. Through the "ascending-descending-lateral" pathway design, it achieves the association and processing of multimodal information. The network architecture is clearly defined and logically sound, with each layer and pathway fulfilling a specific functional role that aligns with the cognitive mechanism of multimodal information integration.

2.Comprehensive Experimental Validation of Feasibility. The paper verifies the feasibility of its method through experiments conducted in both closed and open environments. And in open environments, the experimental results show that the proposed method outperforms other approaches in continuous learning.

**Weaknesses:**

1.Limited Experimental Scope. Current experiments primarily rely on domain-specific datasets and lack validation on more generalized or domain-specific multimodal datasets.
2.Lack of Ablation Studies; Validation of Key Method Components Needed. The paper fails to validate the effectiveness of certain core components through ablation studies. Specific gaps include: No comparison of experimental results under different hyperparameter settings; No analysis of the impact of lateral connections in the network architecture on model performance.
3.The feature selection rules for unimodal data, especially images, lack universality.
4. Inadequate Description of Human-in-the-Loop Conflict Handling and Continuous Learning Details. The descriptions of human-in-the-loop conflict handling and continuous learning processes are insufficiently detailed.

**Questions:**

1.Expansion of Comparative Experiments on Additional Datasets Recommended
2.More ablation studies: like Compare models with and without lateral connections to clarify the role of lateral pathways in feature generalization.
3.Key figures, like Fig. 2, needs to be redrawn.

---

### Official Review · Reviewer_xt4H · 2025-10-30

**Soundness:** 1
**Presentation:** 2
**Contribution:** 2
**Rating:** 2
**Confidence:** 2

**Summary:**

This paper introduces a multimodal learning model for online learning, avoiding catastrophic forgetting and using human-in-the-loop. The model is evaluated on two datasets: Fruit and HomeF.

**Strengths:**

-	Online multimodal learning is a timely and still under-explored topic.

**Weaknesses:**

-	Theoretical assumptions: this paper makes a very strong assumption about the structure of the features given by unimodal pre-trained models. They assume a disentanglement between them (some of them would encode “shape”, others would encode “color” …). This is currently no clear evidence that foundation models such as SAM, CLIP or DINOv3 fulfil this assumption.
-	Validation of the model and benchmarking data: I struggled until the very end of the paper to understand a clear use-case of the proposed model. The issue with the current experimental setup is that it assumes the concepts are unknown to the model before training. Considering the simplicity of the concepts considered (e.g. common fruits or objects), and the fact that the authors used foundation models as pre-trained backbones (trained on web-scale data including most of the “simple” concepts), I believe the model is not learning anything new in its representation. There are no experiments showing that the trained model outperforms simple linear probes on top of the pre-trained backbones in the manuscript. Regarding the datasets used, they are very small (less than 30 objects) and not standard. I would expect a detailed description of these datasets and a justification of why they are more relevant than classical benchmarks in the field (e.g. HowTo100M, AudioSet, AV-Speech, etc…). Finally, no comparison is made with multimodal LLMs, which are currently leading the field.
-	Ablation study: the network introduced on top of the pre-trained backbones is very complex, with a lot of ad-hoc choices (e.g. ascending/descending activation functions for feature neuron, association neuron, multimodal association neuron…). I would expect at least an ablation study on each of the component to justify its utility. Other fusion architectures (more common and simpler such as Transformer) should also be compared.
-	Training and losses: since the authors mentioned “ascending” and “descending” pathways of the information between neurons, I am wondering how the network is trained. This is not mentioned in the original manuscript so I would assume a standard gradient-descent algorithm. If it is the case, I am wondering what is the loss function and how the descending pathway could be concretely implemented when performing the backpropagation.
-	(Minor) The mathematical notations are very uncommon and unclear during the exposition of the model. I would expect less indexes in the variables to clarify the text.

**Questions:**

Please, refer to the weaknesses detailed above for more context on the questions listed below:

- Theoretical assumption: do you have papers or experiments suggesting this assumption holds in foundation models ?
- Training and losses: what is the training strategy ? What loss is optimized ? How the "descending pathway" is implemented in practice ?

---

### Official Review · Reviewer_vTXW · 2025-11-01

**Soundness:** 3
**Presentation:** 2
**Contribution:** 3
**Rating:** 4
**Confidence:** 3

**Summary:**

The paper studies the problem of online multimodal learning and proposes a brain-inspired neural network named OML, which adopts a hierarchical and modular design. Specifically, OML is composed of multiple hierarchical components, including feature neurons, unimodal association neurons (UANs), and multimodal association neurons (MANs). The model integrates ascending, descending, and lateral pathways to facilitate cooperative and interactive learning across modalities in an online setting. Experiments are conducted on the multimodal datasets within the Fruit category and demonstrate that OML achieves superior performance compared to SOTA methods.

**Strengths:**

+ The paper is well-motivated and technically sound, addressing an important problem in online multimodal learning.
+ The proposed OML framework introduces an interesting and intuitive idea inspired by human cognitive mechanisms for learning new multimodal concepts and new associations across multiple modalities in an online manner.
+ The proposed OML framework can detect conflict between the current input and the learnt ones, and check with the users for correctness.
+ The experiments demonstrate the superiority of the proposed method over SOTA methods in the online multimodal learning task.

**Weaknesses:**

- The paper is difficult to follow due to the heavy use of mathematical notations. Many notations are similar or insufficiently explained, and their definitions are scattered throughout the text, forcing readers to frequently refer back and significantly reducing readability.
- One concern is the scalability to a broader range of multimodal tasks. The model adds new concepts when no similar ones are found in memory and no conflicts exist. However, in real-world settings with highly diverse domains and abundant concept variations, it is uncertain how well this mechanism can extend or adapt.
-The experimental validation is limited to a relatively simple dataset (Fruits), which restricts the assessment of OML’s generalization ability to more complex and diverse domains (e.g., animals, furniture, or other domains).
- The robustness to noisy or mispaired data seems to be weak. When neither the visual nor the auditory input is recognized, the model introduces new MANs and pathways, which could lead to incorrect concept formation if the input pairs are mismatched. Moreover, the framework lacks a mechanism to correct or unlearn such erroneous associations once they are established.

**Questions:**

Please address the concerns mentioned in the weaknesses section.

---

### Official Review · Reviewer_W4ES · 2025-11-01

**Soundness:** 2
**Presentation:** 1
**Contribution:** 2
**Rating:** 2
**Confidence:** 4

**Summary:**

This paper introduces OML, a hierarchical neural network designed for online multimodal learning, which allows the model to continuously learn new concepts and associations without forgetting previously learned ones. The network's key features include a "conflict check'' mechanism that identifies discrepancies between new inputs and existing knowledge, and the ability to interactively query a human user for clarification when such conflicts occur. The model also employs a ``reference extraction algorithm'' to determine which specific features, e.g., color vs. shape, a word refers to. Experiments demonstrate that OML avoids catastrophic forgetting in open-ended learning environments and outperforms other methods in tasks requiring precise feature referring and the addition of new modalities.

**Strengths:**

1. This paper studies the problem of multimodal continual learning, which is an important topic for practical deployment consideration.
2. The experimental results demonstrate the effctiveness of the proposed method over included baselines.

**Weaknesses:**

1. The paper's presentation is unsatisfactory, particularly in positioning the proposed work. The authors fail to explicitly identify their approach as continual learning, instead discussing it under the broader category of multimodal learning. This imprecision undermines both the Introduction and Related Work sections, which lack sufficient depth and focus. The inadequate framing suggests the manuscript has not been thoroughly refined and polished.

2. The proposed algorithm exhibits insufficient innovation. The method follows a relatively common approach in continual learning known as Parameter Isolation, such as dynamic architecture expansion. This strategy is established in the continual learning literature but is not clearly acknowledged and discussed for the differences with the proposed method.

3. The experimental evaluation lacks adequate baseline methods for assessing performance in continual learning scenarios. The authors should include a more comprehensive set of state-of-the-art continual learning methods for comparison.

4. The experimental section does not specify whether the reported results represent averages over multiple runs or single-run outcomes. This information is crucial for assessing the stability and reliability of the model's performance, particularly in the continual learning setting where variability across learning sequences can be significant.

5. Another key consideration in continual learning methods is the efficiency of model updates. However, this paper provides no evaluation or comparison of update efficiency, which represents a significant gap in the experimental analysis. Such computational cost are essential metrics for practical continual learning systems and should be thoroughly assessed.

**Questions:**

Please refer to the above weakness for details.

---

### Note · Authors · 2025-11-13

I have read and agree with the venue's withdrawal policy on behalf of myself and my co-authors.